# CoLa: Learning to Interactively Collaborate with Large Language Models

**Abhishek Sharma, Dan Goldwasser**
Department of Computer Science
Purdue University
West Lafayette, IN 47909, USA
{sharm271,dgoldwas}@purdue.edu

## Abstract

LLMs' remarkable ability to tackle a wide range of language tasks opened new opportunities for collaborative human-AI problem solving. LLMs can amplify human capabilities by applying their intuitions and reasoning strategies at scale. We explore whether human guides can be simulated, by generalizing from human demonstrations of guiding an AI system to solve complex language problems. We introduce CoLa, a novel self-guided learning paradigm for training automated *guides* and evaluate it on two QA datasets, a puzzle-solving task, and a constrained text generation task. Our empirical results show that CoLa consistently outperforms competitive approaches across all domains. Moreover, a small-sized trained guide outperforms a strong model like GPT-4 when acting as a guide. We compare the strategies employed by humans and automated guides by conducting a human study on a QA dataset. We show that automated guides outperform humans by adapting their strategies to reasoners' capabilities and conduct qualitative analyses highlighting distinct differences in guiding strategies.

## 1 Introduction

The remarkable ability of Large Language Models (LLMs) to tackle a wide range of text analysis and generation tasks without dedicated task-specific training data (Brown et al., 2020; Raffel et al., 2020; Zhang et al., 2022; Kojima et al., 2022; Yu et al., 2022; Wei et al., 2022; Wang et al., 2022) had a paradigm-changing impact on NLP. Their ability to adapt to new tasks on the fly, react to human feedback, and provide explanations for their decisions has opened the door to more natural interactions between human users and AI systems, in which the two parties can collaborate on complex multi-step tasks, such as supporting social dialogues, collaborative writing, puzzle solving (Lee et al., 2022a;b), data annotation (Kim et al., 2024), and concepts discovery (Viswanathan et al., 2024; Pujari & Goldwasser, 2025). Recent work in both the NLP and HCI communities explored these settings, investigating how to guide and evaluate the collaboration (Amershi et al., 2019; Lee et al., 2022b; Pacheco et al., 2023; Sharma et al., 2023; Collins et al., 2024b).

Given the promise of successful task collaboration between LLM and humans, a natural question is *can LLMs simulate the role of humans in these interactions?* The motivation for investigating this question is two-fold. First, it can equip NLP systems for complex multi-step tasks by capturing the high-level considerations and strategies employed by humans when decomposing such tasks. Second, simulating humans in these interactions can reduce the human effort needed for studying them, by investigating which strategies are effective and how to adapt them to the reasoning capabilities of different LLM partners.

To study this question, we focus on a specific interaction style common in a wide range of collaborative tasks, in which the role of the LLM is a Reasoner making local inferences about the problem, in response to questions and directives from a real or simulated human Guide. The assumption is that neither side knows the correct or optimal response; instead the Guide helps the reasoner identify salient aspects of the input problem through multiple

steps of interactions, with the goal of connecting these intermediate steps to the correct or optimal solution. We present an example in the Fig. 1, where collaboration leads to a *more creative* sentence.

We study this Reasoner-Guide collaboration setting in four tasks, each presenting different challenges. Two question-answering tasks focusing on common-sense (Talmor et al., 2021) and social situations (Sap et al., 2019), where human Guides' intuition for reaching the correct answer is used to lead the Reasoner. A word-puzzle problem, *NYT connections* (Loredo Lopez et al., 2025), challenging for both LLMs and humans, where the Guide helps structure the solution search process and constrained text generation (Lin et al., 2020), where the Guide enriches the generated text while following the constraints.

From a technical perspective, our goal is to learn how to mimic human interaction strategies with the Reasoner model and potentially even improve on these interaction strategies by incorporating domain feedback. We describe our three-step training process in Fig. 2, beginning with recording human interaction with the Reasoner model to identify repeating strategies humans employ when approaching these tasks. We use that seed to create a larger collection of simulated interactions with few-shot prompting using a powerful LLM, GPT-4 OpenAI (2024), on new problem instances. We refer to interaction sequences generated by this approach, that does not include any training, as COLA-PROMPT. We apply COLA-PROMPT to a fraction of the training data, filter out incorrect interactions and train a simulated Guide over the surviving interactions using supervised fine-tuning (SFT). Unlike COLA-PROMPT which uses a powerful LLM, the trained Guide (denoted COLA-SFT) uses a small Language Model (LM), Llama-3B (Meta AI, 2024). Finally, we investigate adapting our simulated Guide to the collaborative task, but instead of memorizing scripted interaction as in COLA-SFT, our next model COLA-RL explores new interaction strategies with the Reasoner, and rewards those leading to better task performance. We study a natural scenario for human interactions, where Guides can improve their strategies even without labels by observing the quality of the Reasoner's responses. We train a reward model (Llama-1B) to predict quality of the interactions, as the Critique and use the PPO RL algorithm (Schulman et al., 2017) to tune the policy of the Guide model. Given the possibility of multiple strategies to collaborate, we propose our next model COLA-RL ENS, by generating multiple beams of interactions between the Guide and Reasoner models and using a novel ensemble-based approach to obtain self-supervision, improving both the Guide and Critique models.

Our experiments show significant task performance improvements in all the tasks when using the COLA framework, using self-supervision performance is improved even further. We show that training a small LM model to act as a Guide, significantly outperforms state-of-the-art LLMs when prompted to act as a Guide. We explored using both a powerful LLM (GPT-4) and a small LM (Llama-3B) as the base Reasoner model, and showed that COLA can help boost the Reasoner's performance in both the cases, highlighted as Step 4 in Fig. 2. We conducted an analysis to investigate the role of the Critique when fine-tuned in COLA-RL ENS develops a better ability to predict the effectiveness of a strategy. We also conducted a qualitative study, comparing the strategies employed by human and several different automated Guides. We compared the usage frequency of different strategies and approximated their effectiveness by measuring the proportion of interactions leading to correct and incorrect outcomes. The results show that humans employ a wider range of strategies compared to simulated Guides, however the most common-strategies are generally similar for human and automated Guides. The same strategies, when employed by humans are more effective compared to COLA-PROMPT, but comparable with COLA-RL.

## 2 Related Work

**Human Interaction**    We draw our inspiration from Human Machine collaboration (Collins et al., 2024a; Lee et al., 2022a;b; Collins et al., 2023; Ippolito et al., 2022; Fan et al., 2019), such that humans interact with machine (language models in our settings), with the objective of collaboration for solving a task. However, the key challenge is effectively scaling human interactions, as obtaining human input is expensive. Furthermore, Lee et al. (2022b) notes that using human interaction can also degrade the task performance. We design CoLa by

building on only handful human interaction examples and effectively scaling them by designing training objectives which model successful collaboration, on a wide range of tasks.

**Prompting**    Prior works  (Brown et al., 2020; Kojima et al., 2022; Wei et al., 2022; Zhang et al., 2022; Yu et al., 2022; Ouyang et al., 2022) have demonstrated impressive capability of LLMs to reason through prompting. However, their nature of prompt sensitivity  (Gao et al., 2021; Zhao et al., 2021) and their reasoning ability dependent on model's size  (Wei et al., 2022; Yu et al., 2024), have motivated works to explore prompt engineering  (Ye et al., 2024; Li et al., 2023) and knowledge transferring from larger LMs to smaller LMs  (Shwartz et al., 2020; Liu et al., 2022). But these approaches when modeling the problem as a single step prompting to generate the task solution in one go, could fail on problems requiring multiple steps of reasoning or continuous refinement to generate an optimal solution. We discuss some works which tackle these two challenges and compare our CoLa framework.

**Problem Decomposition**    Some of the works address the first challenge by decomposing the problem into multiple steps followed by solving each step iteratively to derive the solution  (Zhou et al., 2023; Dua et al., 2022; Wang et al., 2023; Yao et al., 2023a;b), mostly using a single LLM as a module to decompose and solve. Other works use multiple modules to separate the decomposer and solver  (Yin et al., 2024; Xu et al., 2023; Creswell et al., 2022). However, an error during any of these steps could lead into cascading errors. Our work CoLa, provides the flexibility of problem decomposition through interaction where the Guide helps by simplifying the problem for the Reasoner and our training paradigm helps to adapt the Guide based on the Reasoner's problem solving capability and even refine or correct intermediate mistakes through interaction. Furthermore, CoLa supports interactive collaboration between agents – Guide and Reasoner through dialogue acts not plans, drawing this design inspiration from human-machine collaboration.

**Iterative Refinement**    The second line of works addresses the challenge of iteratively refining the LLM's response by either using the same LLM to self-correct or multiple agents to refine the solution  (Madaan et al., 2023; Akyurek et al., 2023; Du et al., 2023). However, this ability of correction is absent in smaller LMs  (Yu et al., 2024). Our framework CoLa refines the LLM's response and pushes task performance for both powerful as well as smaller LM, without touching its parameters.

## 3   CoLa Overview

We introduce CoLa, an interactive framework for Collaborating with Large LMs, by simulating human interaction. CoLa interactively solves a problem through collaboration between two agents - **Guide** and **Reasoner**. The key idea is to simulate Human to collaborate with an LLM. We model collaboration as multiple steps of interactions between Human and LLM, where each interaction is a dialogue exchange between these agents, consisting of text in natural language. The objective of collaboration is to employ a desired *strategy* to guide the LLM to *successfully* solve a given problem. We define the role of LLM as a Reasoner and focus on learning a Guide as a proxy for Human.

We map the sequence of interactions as a conversation between the Guide and Reasoner. Essentially the Guide and Reasoner collaborate through a series of dialogues in natural language to form a conversation. We represent a conversation as $g_1, r_1, g_2, r_2, ..., g_t, r_t$, such that the Guide starts the conversation by generating the first dialogue ($g_1$) which contains the information about the input problem and the immediate next step for the Reasoner. The Reasoner responds by generating the second dialogue ($r_1$). This process is continued until a maximum number of steps ($2t, t \geq 1$), such that $r_t$ corresponds to the problem's solution. Fig. 1 shows an example of a conversation between the Guide and Reasoner for a contained text generation problem.

We assume the Reasoner is a frozen parameters LLM (we evaluate different choices for the base Reasoner) and focus on learning how to provide guidance to it. The first three steps in the Fig. 2 describe the high-level stages of our training protocol. Simply put, the

**User**: Use all 5 words, 'cook', 'grill', 'meat', 'shirt', 'wear', to generate a natural occurring sentence. Be creative but be brief, natural, and coherent.

**Assistant**: I decided to wear my favorite shirt while I cook and grill some delicious meat for dinner.

**Guide**: Set: 'cook', 'grill', 'meat', 'shirt', 'wear'. What could be some scenarios when someone is grilling meat with shirt?

**Reasoner**: Here are a few scenarios: 1. A backyard barbecue where someone is wearing a casual shirt while grilling meat for a gathering of friends and family.
2. A cooking competition where a contestant is grilling meat outdoors, dressed in a shirt that represents their culinary team.
3. A picnic scenario where an individual is grilling meat at a park while wearing a comfortable shirt for the warm weather.
4. A father grilling meat on the patio for his children while wearing a funny shirt that reflects his personality.
Do any of these scenarios resonate with you, or would you like to explore more ideas?

**Guide**: I like the father example. Also add the situation when the father stains his shirt?

**Reasoner**: Sure! In this scenario, the father could be grilling meat for his children on the patio while wearing a funny shirt. As he flips the meat, he accidentally spills some marinade or sauce on his shirt, leaving a noticeable stain. This could lead to a humorous moment where the kids tease him about the stain, making the situation light-hearted and memorable. Would you like to incorporate this detail into a final sentence, or do you have more ideas to add?

**Guide**: Ok, generate a sentence using this idea with all the 5 words.

**Reasoner**: The dad decided to cook and grill meat for his kids while wearing a funny shirt, but he ended up staining it with barbecue sauce.

Figure 1: *Comparing instruction prompting (upper) and CoLa (lower) on CommonGen.* CoLa uses Llama Guide trained using RL (CoLa-RL) and GPT4 Reasoner

pipeline begins with collecting human demonstrations for our four tasks (Sec. 3.1), next we automatically create high quality interaction data using human demonstration by prompting GPT-4 and filtering interactions that lead to incorrect decisions (Sec. 3.2). Next, we train the Guide on this data, initially using direct fine-tuning and then using RL over unlabeled data (Sec. 3.3). We provide details in the following sub-sections for different versions of the CoLa framework.

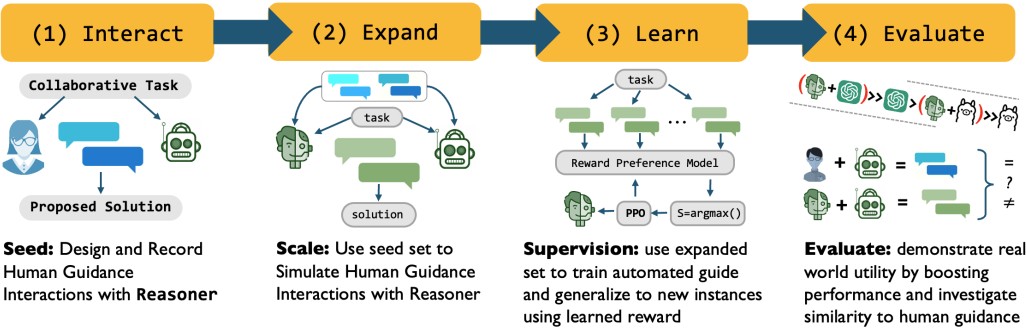

Figure 2: *CoLa framework stages.* (1) Generate seed by collecting Human-GPT-4 interactions, (2) generate interaction training data using the seed demonstrations (3) train Guide and Critique models using SFT, and use trained Critique model for RL training on unlabeled instances.(4) Evaluate CoLa variants and compare to human strategies.

## 3.1 Interacting

We sample a handful number of examples for each domain ($< 10$) and ask a human to interact with GPT-4[1] to solve each problem. We share the annotation instructions and

---
[1]We use OpenAI GPT [Playground](#)

the complete process of sampling the selected examples in Appendix A.1. The human collaborator is unaware of the true solution of these problems, making this process as a true reflection of our objective. This interaction step results into forming a small seed set of conversations.

## 3.2 Expanding

We sample a small portion of examples from the training set for each domain, statistics in Tab. 4. We use GPT-4 as a simulated Guide to interact with the Reasoner. We use the seed set of conversations (Sec. 3.1), as few-shot exemplars to instruct prompt (Ouyang et al., 2022) GPT-4 as a Guide. We use the same set of exemplars to prompt the Reasoner. We provide all these prompts in App. A.2. We refer to this design as COLA-PROMPT which doesn't fine-tune any language model but relies on in context learning (Brown et al., 2020).

## 3.3 Learning

**SFT Design**  We start with a pre-trained language model Llama-3B (Meta AI, 2024) as a starting point of the Guide model. We process the generated conversations and filter into *successful* and *unsuccessful* categories. A conversation is successful if it leads to the correct solution, more specifically $r_t$ matches to the ground truth label of the example or satisfies all the constraints in the case of the constrained text generation task.
We then create the training set for the Guide, by selecting each successful conversation and unfolding it into tuples of generated dialogues. For eg, a conversation $g_1, r_1, ..., g_t, r_t$ will result into $t$ training examples $\{(g_1), (g_1, r_1; g_2), ..., (g_1, r_1, ..., r_{t-1}; g_t)\}^2$.
We use decoder-only model and use supervised fine-tuning (SFT) to train on the target dialogues. We refer to this model as COLA-SFT.

**RL Design**  We explore learning without an access to the ground truth label. We draw inspiration from Reinforcement Learning (RL) (Sutton & Barto, 2018) treating the Guide's parameters as a tunable policy to generate a dialogue as an action ($g_i$), based on the context as the state ($g_1, r_1, ..., r_{i-1}$). We compute the reward of each action by training a reward preference model - Critique. The role of the Critique is to measure the likelihood of an interaction leading to the successful collaboration.
Similar to the SFT stage, we process the generated conversations to prepare the training set for the Critique model in the following way. We unfold all the conversations (successful and unsuccessful) into tuples of $t$ training examples, $\{(g_1; s_1), (g_1, r_1, g_2; s_2), ..., (g_1, r_1, g_2, ..., r_t, g_t; s_t)\}$, where $s_i$ is the label representing successful or unsuccessful in natural language text. If a conversation is successful then each $s_i$ is successful and vice versa. Similar to SFT we fine-tune the Critique to train on these labels. Using the Critique generation, we define the reward function for a given interaction ($g_1, r_1, ..., r_{i-1}, g_i$) as an integer value, $w_i$:

$$w_i = \begin{cases} +1, & \text{if } s_i = \text{successful} \\ -1, & \text{otherwise} \end{cases} \tag{1}$$

We use this value to train the Guide using the PPO algorithm (Schulman et al., 2017). We refer to this model as COLA-RL.

**Ensemble Design**  We further explore improving the Critique model to continue improving the interaction process. We design this iterative improvement loop using the technique of ensemble. We draw the inspiration from the EM (expectation-maximization) algorithm, such that in the E-step we generate multiple beams of conversations between the Guide and Reasoner, and score each conversation based on the reward generated by the Critique model, to compute the expected *posterior* label using the reward score as the weighted ensemble. Whereas, in the M-step we use the previously generated conversations to update the parameters of the Critique model, based on the reward score of the conversations.

---

[2]We always initialize the Guide's input with the problem description which we omit in the notation for compactness.

During the start of the conversation we prompt the Reasoner to generate top-$K$ responses. Each response is then provided to the Guide and the interactions are run in-parallel, creating $K$ conversations thereafter. For a $k^{\text{th}}$ beam of conversation, we compute the reward for each dialogue generated by the Guide and perform the PPO update step.

We also update the Critique model by fine-tuning on the generated interactions. In order to construct the training set labels for Critique we use weighted ensemble to estimate (E-step) the input's label as a proxy for the ground truth label.

We define the weight as the average reward value for the entire conversation, $\hat{w}^k$:

$$\hat{w}^k = \frac{\sum_{i=1}^{t^k} w_i^k}{t^k} \tag{2}$$

where $t^k$ is the number of Guide interactions for the $k^{\text{th}}$ conversation, and $w_i^k$ is computed using Eq. 1. The intuition is that if $\hat{w}^k$ is close to 1 then then almost all the Guide interactions for the $k^{\text{th}}$ beam will lead to the successful collaboration.

We extract the input label for a conversation, $y^k$ from $r_t^k$ (Reasoner's last response) where $y$ maps to the label choice of the task, eg, $y$ would be an option for the question answering task. Now we estimate the final label using $\arg\max$ over weighted voting of each beam, $y$:

$$y = \arg\max_y \sum_k \hat{w}^k y^k \tag{3}$$

Now that we have the estimate of the input label we compare each beam's label, $y^k$ against $y$ to determine if the $k^{\text{th}}$ conversation is successful or unsuccessful. This allows us to construct the training examples for the Critique ($s_i^k$) and update the parameters by performing SFT (M-step). The rationale is that by generating $K$-fold conversations for each problem, results in multiple rationales as solutions to the original problem, providing a rich training data for the Critique model. This yields a better reward signal for the Guide model for the RL step, such that both the models could improve through this iterative process.

## 4 Experimental Evaluation

### 4.1 Setup

**Tasks** We benchmark on four domains - two question-answering (QA) tasks, SocialIQA (Sap et al., 2019) and CSQA2 (Talmor et al., 2021), a puzzle task, NYT Connections (Loredo Lopez et al., 2025), and a constrained text generation task, CommonGen (Lin et al., 2020). We provide the dataset statistics in Tab. 4.

**Baselines** We compare CoLa against chain-of-thought style prompting approaches such as zero-CoT (Kojima et al., 2022) and CoT (Wei et al., 2022) for the QA domains, and instruction prompting (instruct) (Ouyang et al., 2022) for Connections and CommonGen. We also compare CoLa against Self-Refine (refine) (Madaan et al., 2023), where the language model has the ability to self-critique and improve its generation iteratively, and Multi Agent Debate (MAD) (Du et al., 2023), where multiple-agents engage in debate discussion to refine each others responses to arrive at a consensus.

For a fair comparison we use the same number of steps for refine and number of debate rounds for MAD, as the maximum number of interaction steps for CoLa for each domain, statistics Tab. 4. We use two agents for the MAD framework for a fair comparison against our setup of two models - Guide and Reasoner. For MAD, we use the Agent 1's response as the final response.

We also compare CoLa against fully-supervised (sup) approach in which a model as the same size as the Guide model (Llama-3B) is trained with the same amount of labeled used to train COLA-SFT.

**Model Choices** We benchmark two pre-trained language models - GPT-4-o-mini (GPT) (OpenAI, 2024) and Llama 3.2 3B *instruct-chat* (Llama) (Meta AI, 2024) as our choice for the Reasoner models. Similarly we choose GPT and Llama 3.2 3B *instruct-chat* (Llama) as our

Guide models, where learning is performed for the Llama Guide model. We use Llama 3.2 1B *instruct-chat* as our Critique model. We only evaluate COLA-RL ENS for the QA domains, as the other domains have a larger space of label choices, leaving it as a future work.

**Implementation**    We implemented all the learning algorithms using TRL  (von Werra et al., 2020) in Huggingface  (Wolf et al., 2020). We use temperature  (Holtzman et al., 2020) for Guide as 0.2 and for Reasoner as 0.1 for all our generations. For COLA-RL ENS, we use $K = 5$ and $K = 3$ as the top-$K$ for SocialIQA and CSQA2 respectively.

**Evaluation**    For the QA and Connections tasks, we evaluate the LLM's response against the ground truth label of the input examples. For each system we prompt the underlying Reasoning LLM to generate the final answer based on it's reasoning or final reasoning step (for CoLa, MAD, and refine), similar to  (Kojima et al., 2022). For the QA tasks, we compare the final answer against the ground truth option choice and compute the average percent of accuracy across the test split (Acc, ideal value: 100).

For Connections the final response contains groupings of the puzzle words, and we use the following 3 metrics:

1. Acc: measures the full correctness of each puzzle's solution and we compute the average percent across the test split (ideal value: 100).

2. G-Acc: measures the number of groups (out of 4) correctly discovered (entirely) and we report the average across the test split (ideal value: 4).

3. Purity: measures how many words in each group are correctly clustered, we compute the percent value for each puzzle and average across the entire test split (ideal value: 100). Note that a completely randomized solution will always have a Purity of at least 25%.

For CommonGen we have human written reference for the test split. We use GPT-4 as a judge  (Zheng et al., 2023) to compare the *coherence* and *creativity* of human reference vs LLM generated sentences. We prompt GPT-4 to pick a *winner* (or *tie*). We report the Win+Tie average percent across the test split (Acc, ideal value: 100). We also measure how many words from the givent input set were used while generating the sentence as Coverage and report the average percent.

## 4.2    Results

**QA Results**    We show the results for the QA tasks in Tab.  1. We notice that COLA-PROMPT performs competitive against the baselines except under-performs against MAD. This overall shows the importance of multi-step interactions against prompting techniques. Furthermore, stronger Guide and Reasoner contribute to higher performance. We notice that when a stronger Reasoner is paired with a smaller Guide model (Llama ⊗ GPT COLA-PROMPT), it significantly hurts the performance. However, with training this completely changes the result and the same Guide (after SFT) out-performs the strong Guide (when prompted), Llama ⊗ GPT COLA-SFT vs GPT ⊗ GPT COLA-PROMPT. Additionally, COLA-SFT outperforms all the baselines including MAD.
We also notice that a trained Guide (COLA-SFT) boosts performance for both a stronger Reasoner (GPT) as well as smaller Reasoner (Llama), interestingly benefiting smaller Reasoner by a larger margin. We suspect this could be due to the reason that baseline performance on these QA tasks is already *high*, therefore less room for improvement.
We also show for Llama Reasoner that COLA-SFT performs better than the fully-supervised model, with the same amount of labeled data.
With RL, the performance continue to increase, COLA-RL ENS achieving the best results for for GPT and LLama Reasoners, demonstrating an effective technique for interaction.

**NYT Connections**    We show the results for the puzzle task in Tab. 2. The first observation is the difficulty of the task showing state-of-the-art GPT-4 model struggling on the task and Llama performing quite poorly. We notice that COLA-PROMPT out-performs all the

| Model | SocialIQA | CSQA2 | Model | SocialIQA | CSQA2 |
|---|---|---|---|---|---|
| GPT (zero-CoT) | 79.1 | 82.1 | Llama (zero-CoT) | 68.2 | 63.8 |
| GPT (CoT) | 79.8 | 80.8 | Llama (CoT) | 69.1 | 61.2 |
| GPT (refine) | 80.6 | 81.9 | Llama (refine) | 65.7 | 62.7 |
| GPT (MAD) | 82.0 | 83.4 | Llama (MAD) | 73.1 | 64.9 |
| GPT ⊗ GPT COLA-PROMPT | 80.4 | 82.0 | Llama ⊗ Llama COLA-PROMPT | 71.0 | 62.1 |
| Llama ⊗ GPT COLA-PROMPT | 76.5 | 78.0 | Llama ⊗ Llama COLA-SFT | 75.6 | 68.1 |
| Llama ⊗ GPT COLA-SFT | 83.1 | 84.6 | Llama ⊗ Llama COLA-RL | 77.0 | 70.2 |
| Llama ⊗ GPT COLA-RL | 85.6 | 86.1 | Llama ⊗ Llama COLA-RL ENS | **77.7** | **71.9** |
| Llama ⊗ GPT COLA-RL ENS | **86.8** | **88.3** | Llama (sup) | 73.8 | 64.8 |

Table 1: Evaluating on the QA tasks. CoLa framework for Guide ⊗ Reasoner.

baselines, demonstrating the usefulness of step by step interaction for effective collaboration. We observe that with SFT the performance is significantly improved for COLA-SFT models. For GPT as a Reasoner, Llama ⊗ GPT COLA-SFT out-performs GPT ⊗ COLA-PROMPT, which is a quite motivating result because Llama as an individual model (non-CoLa systems) performs quite worse when compared against the performance of GPT, but post-training the same model as a Guide out-performs GPT, demonstrating its capability to be a successful collaborator as a Guide even in the domain where it struggles as an individual solver. Furthermore, with a trained Guide, Llama even as a Reasoner out-performs the powerful GPT as a Reasoner (Llama ⊗ Llama COLA-SFT vs GPT ⊗ GPT COLA-PROMPT), making a point that even with a smaller LM a trained Guide can outperform a powerful system. Additionally, Llama as a Reasoner as part of the COLA-PROMPT model, significantly out-performs the fully supervised model, proving the usefulness of interaction as opposed to only labeled data.

COLA-SFT boosts performance for both the Reasoners, but in this domain we observe that it benefits the stronger Reasoner by a larger margin, proving stronger LLM more effective for a complex domain when paired with a trained collaborator.

Finally we notice that COLA-RL further increase the performance, achieving the best results for both the Reasoners and more than twice accurate as compared to the best baseline (MAD).

**CommonGen**   We show the results for the constrained text generation task in Tab. 2. We observe that satisfying constraint is quite easy (Coverage), however, the challenge is to generate a more creative and coherent sentence when compared against human reference. We notice that COLA-PROMPT out-performs all the baselines except refine. Post training COLA-SFT out-performs the refine model, the difference of improvement further increases after RL phase COLA-RL. We notice that stronger Reasoner model GPT benefits more through collaboration, which we suspect is due to the fact that output of smaller LMs is more challenging to improve (Yu et al., 2024).

As a note when GPT is used as the prompting or Reasoning model we see the Win+Tie rate over 50, meaning GPT generated sentences are ranked higher than human references, based on the judge. We also manually compared generated sentences for few randomly sampled examples to verify annotation agreement with GPT-4. We observe that this reason is due to the prompt construction, as the LLM is asked to generate a *creative* sentence, which was not the case when humans annotated this dataset (Lin et al., 2020).

### 4.3   Critique Analysis

Given the impressive performance of the CoLa models we perform an analysis of the role of Critique. For the QA domains, we obtain the generated response of Llama (CoT), Llama ⊗

| Model | Connections | | | CommonGen | |
|---|---|---|---|---|---|
| | Acc | G-Acc | Purity | Win+Tie | Coverage |
| GPT (instruct) | 11.2 | 1.1 | 67.8 | 51.7 | 98.3 |
| GPT (refine) | 13.8 | 1.3 | 68.7 | 59.2 | 99.3 |
| GPT (MAD) | 15.1 | 1.4 | 70.1 | 53.7 | 98.6 |
| GPT ⊗ GPT COLA-PROMPT | 15.7 | 1.4 | 71.4 | 54.1 | 98.7 |
| Llama ⊗ GPT COLA-PROMPT | 10.8 | 1.0 | 62.9 | 46.4 | 98.1 |
| Llama ⊗ GPT COLA-SFT | 27.0 | 2.5 | 87.6 | 60.1 | 99.8 |
| Llama ⊗ GPT COLA-RL | **32.4** | **2.7** | **89.8** | **64.7** | **99.9** |
| Llama (instruct) | 4.7 | 0.6 | 49.3 | 22.4 | 90.6 |
| Llama (refine) | 5.8 | 0.7 | 54.6 | 27.1 | 95.8 |
| Llama (MAD) | 8.7 | 0.9 | 61.7 | 24.8 | 92.3 |
| Llama ⊗ Llama COLA-PROMPT | 7.1 | 0.8 | 59.8 | 24.3 | 92.7 |
| Llama ⊗ Llama COLA-SFT | 18.1 | 1.8 | 72.8 | 29.6 | 97.2 |
| Llama ⊗ Llama COLA-RL | 20.2 | 1.9 | 74.1 | 32.4 | 98.9 |
| Llama (sup) | 11.0 | 1.0 | 64.1 | 28.0 | 96.1 |

Table 2: Evaluation on **NYT Connections** (Acc, G-Acc, Purity) and **CommonGen** (Win+Tie, Coverage) using CoLa framework for Guide ⊗ Reasoner.

Llama COLA-PROMPT, and Llama ⊗ Llama COLA-RL ENS, on the test set. We then apply a Critique model to judge if the generated response is correct. We test three types of Critique model - critique as a pre-trained Llama-1B, and our trained Critique models obtained post SFT (SFT critique) and post RL ensemble (RL ens critique).

For the positive class i.e. the true answer choice of the test set examples, we measure the ability of different critique models to predict if a response corresponds to the true labels or not. We report the precision (P), recall (R), F scores on the positive class in Tab. 3.

We first observe that model producing better quality response COLA-RL ENS, mostly has the highest F scores for any type of critique. This is a promising result because the critique not trained on any CoLa conversation, is still able to judge the correctness through the quality of the response. This also explains higher recall by trading precision, as the critique mostly judges CoLa responses as correct.

Second as the Critique is fine-tuned on CoLa conversations, it develops a better ability to judge the responses against the ground truth, with performance increasing for COLA-RL ENS model.

| Model + Critique | SocialIQA | | | CSQA2 | | |
|---|---|---|---|---|---|---|
| | P | R | F1 | P | R | F1 |
| Llama + critique | 0.91 | 0.85 | 0.88 | 0.86 | 0.79 | 0.82 |
| Llama ⊗ Llama COLA-PROMPT + critique | 0.77 | 0.94 | 0.85 | 0.76 | 0.94 | 0.84 |
| Llama ⊗ Llama COLA-RL ENS + critique | 0.81 | 0.93 | 0.86 | 0.85 | 0.88 | 0.86 |
| Llama ⊗ Llama COLA-PROMPT + SFT critique | 0.84 | 0.87 | 0.85 | 0.89 | 0.95 | 0.92 |
| Llama ⊗ Llama COLA-RL ENS + SFT critique | 0.87 | 0.96 | 0.91 | 0.86 | **0.98** | 0.92 |
| Llama ⊗ Llama COLA-PROMPT + RL ens critique | 0.88 | 0.89 | 0.88 | 0.93 | 0.89 | 0.91 |
| Llama ⊗ Llama COLA-RL ENS + RL ens critique | **0.95** | **0.97** | **0.96** | **0.95** | 0.95 | **0.95** |

Table 3: Critique Success. CoLa - Guide ⊗ Reasoner

## 5 Human Interaction

We conduct a study to understand how effective is our CoLa simulated Guide when compared to actual human guides. Our primary goal is to observe the similarities and differences between the interactions of a human vs a simulated human with an LLM. For this we propose an interactive experiment between Guide and Reasoner. We fix the Reasoner model to Llama-3B, and the Guide could either be a human or CoLa Guide. We provide the experiment details in A.3. We analyze the interactions of humans with the Reasoner model and compare against the CoLa interactions.

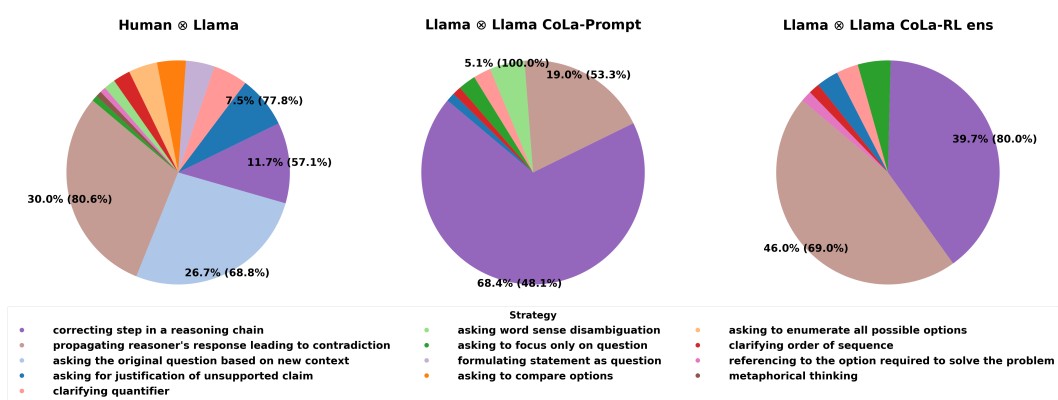

Figure 3: CSQA2 Interactions Strategy Pie Chart

**Human vs. CoLa: who is a better guide?** We observe the quantitative results in Tab. 5. The results positively demonstrate how training a Guide is more effective at guidance when compared against humans. However, these QA tasks itself are not challenging for humans, so we further investigate the interactions to discover the reason for humans not out-performing the CoLa guides. We observe that humans mostly have an idea of the correct answer, however, they interact using complex reasoning strategies on which the Reasoner finds challenging to adapt. Whereas, the CoLa guides were specifically trained with the Reasoner model, making the adaptation process easier.

**Do Humans and CoLa guide *differently*?** Our next experiment compares the fine grain strategies employed by humans vs the CoLa guides. We select the CSQA2 task and qualitatively analyze the strategies. We sample few interactions and come up with a strategy for each sampled interaction. We then use these as few-shot exemplars to generate the remaining strategies for each interaction, by prompting GPT4-o-mini (OpenAI, 2024).

We show the distribution of these strategies for each Guide in Fig 3 and Tab 6 – with the frequency of a strategy along with its correctness i.e. how often does a given interaction's strategy lead to the correct answer at the end of the conversation. We observe that humans employ a wide range of strategies following a long tail distribution, with most of the common ones present across CoLa models as well. A common strategy shared by all the Guides is correcting intermediate reasoning steps through the interactions (`correcting step in a reasoning chain`). We show that COLA-PROMPT is ineffective by over-generalizing this strategy, whereas COLA-RL ENS effectively employs this strategy, even better than humans. A key observation is that a common strategy used by humans while guiding Reasoner for QA is first providing some information and then asking the original question based on this new information (`asking the original question based on new context`), is completely absent in CoLa models. We believe this as a unique cognitive level understanding of the needs of a problem and assessing when enough information is available to answer the question; proposing this a motivation for future work to improve collaboration in automated systems.

## 6 Conclusion

Drawing inspiration from human machine collaboration we propose CoLa, a framework to simulate human guidance for LLMs with the goal of solving a wide range of language tasks. CoLa provides guidance to LLMs by generating dialogues and engaging in a multi-turn conversation leading to the task solution. We discuss various training approaches to learn to simulate the human guidance, and propose a novel self-supervision learning paradigm based on Reinforcement Learning, which out-performs competitive baselines. We also conduct a human analysis to study the differences between human guidance and CoLa guidance, observing that humans employ wide range of guidance strategies, whereas CoLa when trained can out-perform human guidance and perform some of the guidance strategies better than humans.

## Acknowledgments

We are very grateful to all the members of the Purdue NLP lab, for their help with the annotations and valuable suggestions for CoLa. We also thank the reviewers for their feedback for improving the paper. This paper was partially supported by an NSF CAREER award IIS-2048001.

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

# A Appendix

## A.1 Seed Set Conversation Gneration

**SocialIQA** SocialIQA is a Question Answering task, such that for a given *Situation*, the reader has to answer a multiple-choice question using *social commonsense*. We observe that some examples have answer choices which require nuanced social commonsense reasoning. Therefore the objective for conducting human interaction is to guide the Reasoner model why certain choice is more appropriate than the other choices. We want to have broad representative examples as part of the few example seed set. Therefore, we first run the Reasoner model on a train subset (around 100 examples) by prompting it in a zero-CoT setting. In this way we collect examples which the Reasoner model answers correctly and incorrectly, using the ground-truth labels. For the examples which the Reasoner model failed to solve in the zero-CoT setting, we sample a small set (around 10) examples for the human interaction process.

We start the interaction process by presenting the Reasoner model with the example and prompting it in zero-CoT setting as part of the default initial Guide prompt ($g_1$), and based on the response ($r_1$), the Human acts as a Guide by proving a feedback ($g_2$). If the Guide *believes* that the reasoning and the answer in the initial attempt of the Reasoner model is correct, then they provide a *positive* feedback and the interaction process terminates, generating $g_1, r_1, g_2$ as a short conversation. However, if they feel that the reasoning or the answer has some error then they provide a feedback ($g_2$) with the objective of addressing that error. The Reasoner takes the Guide feedback into account to re-generate the solution ($r_2$). Based on the new response, the Guide again provides a feedback ($g_3$). This interaction process continues until $g_i$ is a positive feedback or the conversation reaches a maximum depth ($t \to g_t$).

During this entire collaboration process we collect interactive conversations between Human acting as the Guide and GPT4m acting as the Reasoner. This generates conversations as $g_1, r_1, ..., r_t, g_t$ such that $t \geq 1$. We finally prompt the Reasoner to generate the answer choice based on the conversation to produce $g_1, r_1, ..., r_t, g_t, r_t$. We then filter the conversations by only keeping the ones which arrived to the final solution, by comparing $r_t$ with the ground label.

**Common Sense Question Answering (CSQA2)** CSQA2 is a question answering task, such that given a Question, the reader has to use *commonsense* to answer it in *yes* or *no*. Some of the questions require *retrieval*, *arithmetic operations*, *ordering between actions*, etc. This nature of problem might require complex reasoning skills, therefore the objective of collaboration would be to provide feedback based on the skill(s) required to correctly answer a question. We conduct the interaction process similar to the above SocialIQA domain, to generate the set of conversations.

**Connections** Connections is a word puzzle game featured in NYT. We crawl the puzzle from this link and use the unlimited archive as our training set containing, 500 puzzles, whereas select NYT featured puzzles from the dates: June-6-2024 to Nov-26-2024 to collect 321 puzzles as part of test.

Each puzzle contains 16 words and the task is to cluster all the words into 4 different clusters, with each cluster containing unique 4 words. The clustering is performed based on a common theme shared by a group of 4 words. A theme could be simple, for eg, words sharing theme of "Lead Singers of '70s Rock Bands: Ferry, Mercury, Nicks, Plant" or as abstract as words being "Palindrome". We observe this is a challenging task for language models, even for powerful models like GPT4m. We model this task as a clustering task such that the Reasoner should attempt to step-by-step cluster words and the Guide should provide hint(s) or feedback for each step such that all the words could be clustered correctly.

To design the interaction process for this task on a given puzzle with random order of 16 words, the first two prompts are fixed. The first prompt ($g_1$) asks the Reasoner to enumerate all possible unique word pairs for the 16 words candidate, generating $\frac{16 \times 15}{2} = 120$ word pairs as the response ($r_1$). The second prompt ($g_2$) asks to find all the possible connections between each pair.

Based on the last response of all the possible word connections ($r_2$), Guide starts the interaction process by providing hint(s) using the common themes to start clustering the words. This is performed iteratively such that the Guide provides hint at each step ($g_i$) with the objective of producing correct clusters satisfying the puzzle requirements. Finally when the Guide observes that the Reasoner has clustered all the words in a given response ($r_{t-1}$), they prompt the Reasoner with a fixed final question ($g_t$). This ensures that the Reasoner generates the puzzle solution ($r_t$) which can be easily verified for correctness.

We collect a set of interactions and only select the conversations which produced the correct final answer. We observe despite language models struggling on this complex task and even humans facing challenge while solving this task, using this interaction process is quite effective, meaning most of the collected conversations resulted into full correctness. So we only select one conversation as an example for the collaboration process for this domain. Furthermore the length of conversation for this domain is much longer than the length of conversation for the other domains, therefore, we limit to selecting only one conversation for Connections, to budget the computational cost.

**CommonGen** CommonGen is a constrained text generation task, such that given a set of words one must generate a *coherent* sentence using all the words. This domain observes that language models struggle to generate coherent sentences when compared against the ability of humans to generate sentences. Therefore, the objective for collaboration in this domain would be to ensure Reasoner generates coherent sentences satisfying the constraint of utilizing all the words. Furthermore, we also focus on another objective of crafting these sentences as *creative* while being coherent at the same time.

For this task, given an input example with the set of words, the Guide first interacts with the Reasoner by asking for the *scenarios* where these words will *naturally* occur, with a crafted prompt ($g_1$) for the specific input. The Reasoner responds by suggesting the possible scenarios ($r_2$). The Guide then selects a specific scenario and also suggests creative additions to the given scenario ($g_2$). In this interactive process, the Reasoner finally comes with a creative scenario ($r_{t-1}$) utilizing the input words. The Guide finally asks ($g_t$) the Reasoner to produce the final sentence ($g_t$).

| Data | SFT Train | RL Train | Test | Max Steps ($t$) |
|---|---|---|---|---|
| SocialIQA | 3k | 8k | standard test | 3 |
| CSQA2 | 2k | 4k | standard dev | 3 |
| NYT A.1 | 250 | 250 | 321 | 7 |
| CommonGen | 700 | 700 | standard test | 5 |

Table 4: Dataset details with training and evaluation splits.

## A.2 Prompts

---

**SocialIQA**

**Context:** Sydney kept Lee apart from family after gaining custody through a long divorce settlement.
**Question:** Why did Sydney do this?
A. Spite the family
B. Build a new life
C. Move on from the divorce
**Let's think step by step.**
**Assistant:** To analyze why Sydney might have kept Lee apart from the family after gaining custody, we can consider the motivations behind such a decision.
A. "Spite the family" suggests that Sydney's actions were driven by a negative desire to hurt the family, which could be plausible given the contentious nature of divorce, but it doesn't necessarily imply a constructive outcome.
B. "Build a new life" implies that Sydney wants to establish independence and create a new environment for Lee. This can be a common motivation after a divorce, where one seeks to redefine family structures and personal circumstances.
C. "Move on from the divorce" indicates that keeping Lee apart could be seen as a step toward healing and creating closure, allowing both Sydney and Lee to transition emotionally.
Considering the context of a long divorce settlement, it seems more likely that Sydney's actions are motivated by B or C rather than simply out of spite. However, "building a new life" (B) seems to capture the proactive nature of creating a fresh start. Given this reasoning, I would select B. build a new life.
User: But in order to build a new life isn't the first step to move on?
Assistant: You bring up a good point. Moving on from the divorce is indeed an essential first step in order to successfully build a new life. If Sydney is keeping Lee apart to move on, it could be part of the process to eventually create that new life. Therefore, considering your feedback, "move on from the divorce" (C) could be interpreted as a foundational step leading to the ultimate goal of building a new life (B). It emphasizes the emotional transition before tangible changes take place. With this in mind, I would revise my choice to C. move on from the divorce.
User: What is the final answer A, B, or C? Provide the answer in
Answer. Eg,
AnswerA.
AnswerC
**Context:** Jordan wasted a day playing video games at home.
**Question:** What will Jordan want to do next?
A. Play more
B. Sleep
C. Eat
**Let's think step by step.**
**Assistant:** Given that Jordan wasted a day playing video games, it's likely that he might want to continue that activity. Therefore, the most logical choice is: A. play more.
User: But the context mentions he wasted the entire day so wouldn't he now consider resting given it might be night?
Assistant: That's a valid point. If Jordan spent the whole day playing video games, he might feel tired by the end of the day, which would lead him to want to rest. This suggests that: B. sleep could indeed be the correct choice. Thank you for the clarification!
User: What is the final answer A, B, or C? Provide the answer in
Answer. Eg,
AnswerA.
AnswerB

---

---

**CSQA2**

Question: Egypt comes before causes alphabetically.
Let's think step by step.
Assistant: To determine if "Egypt" comes before "causes" alphabetically, we compare the first letters: 'E' comes before 'c' in the alphabet. Therefore, "Egypt" does come before "causes." So the answer is yes.
User: But 'c' comes before 'E' alphabetically.
Assistant: You are correct. 'C' comes before 'E' in the alphabet. Therefore, "Egypt" does not come before "causes." The answer is no.
User: What is the final answer yes or no? Provide the answer in \Answer{}. Eg, \Answer{yes}.
Assistant: \Answer{no}.
Question: Touch screens became part of Xerox's commercially available products in the 1980s.
Let's think step by step.
Assistant: Yes, Xerox developed early touch screen technology in the 1970s, and by the 1980s, they incorporated it into their products, making it commercially available.
User: Maybe Xerox developed early touch screen products but were they also first to commercialize them?
Assistant: No, Xerox was not the first to commercialize touch screen technology, although they developed it. Other companies, like IBM, also worked on touch screen technology during that era and brought it to market.
User: What is the final answer yes or no? Provide the answer in \Answer{}. Eg, \Answer{yes}.
Assistant: \Answer{no}.

---

**CommonGen**

You will be given a set of words and asked to generate a sentence using all the words. You should generate a sentence that is coherent and natural.

A User will guide you step by step to form ideas to generate a natural sentence using the given words. Once all the ideas are ready, the User will ask you to generate the sentence. You should only then generate the sentence.

Only generate one sentence using all the words. Be creative but also make sure the sentence is brief, coherent, and natural.

**Example 1: Set:** 'desk', 'keyboard', 'laptop', 'monitor', 'mouse'

**User:** Think of an underlying setting where these devices will naturally occur.

**Assistant:** A common setting where a desk, keyboard, laptop, monitor, and mouse would naturally occur is a home office or a workspace. This environment typically includes a work desk where a person uses a laptop connected to a monitor, with a keyboard and mouse for input. Would you like to explore more ideas related to this setting?

**User:** Ok, let's focus on an office setting. Now be creative and generate the sentence using all these 5 words.

**Assistant:** In the bustling office, I sat at my desk, typing away on the keyboard of my laptop, while glancing at the monitor and using my mouse to navigate through the endless streams of information.

**Example 2: Set:** 'carrot', 'corn', 'meat', 'plate', 'potato'

**User:** Think of an event where these food items will be present.

**Assistant:** A family barbecue or picnic would be a great event where these food items could be present.

**User:** How about Thanksgiving?

**Assistant:** Thanksgiving is an excellent choice! It often features a variety of dishes, including roasted meats and side dishes like mashed potatoes, corn, and even carrot-based dishes.

**User:** Ok, generate a sentence focusing on a Thanksgiving celebration with all these 5 food items.

**Assistant:** During the Thanksgiving celebration, the table was beautifully set with a plate piled high with succulent meat, creamy mashed potatoes, sweet corn, and glazed carrots, making it a feast to remember.

---

### A.3 Human Analysis

We sample a few examples from the test split of the QA tasks, around 200 examples from CSQA2 and 50 examples from SocialIQA, and ask humans to interact with LLama-3B in a similar style as CoLa. We share the same instructions and guidelines as mentioned in App A.1 and analyze their responses. Tab. 5 shows the performance of different interactive systems when using different Guide with the same Reasoner model. We also compute the average number of interaction rounds (Rounds), and relevance (Rel) of each interaction by prompting GPT-4 with the contextualized conversation and ground truth label to ask if the given Guide interaction is *relevant* or not. We observe that COLA-SFT and COLA-RL ENS out-perform humans when acting as Guide, a positive result of learning to interact effectively.

### A.4 Limitations

CoLa is designed for solving any language based task, however we note the importance of domain specific training to boost the task performance, over general purpose prompting or multi-agent approaches. Although the Reasoner model is general purpose and not fine-tuned, however, we require the Guide model to be trained in order to effectively pair with the Reasoner model as a collaborative partner. Furthermore, the starting point for designing the guidance steps is obtained through human interactions – forming the initial step for interactive conversation templates, although we only require a handful of these human

| Model | SocialIQA | | | CSQA2 | | |
|---|---|---|---|---|---|---|
| | Acc | Rounds | Rel | Acc | Rounds | Rel |
| Llama (zero-CoT) | 50.0 | - | - | 27.1 | - | - |
| Human ⊗ Llama | 58.3 | 1.4 | 84.8 | 65.5 | 1.8 | 88.7 |
| Llama ⊗ Llama CoLA-Prompt | 69.1 | 1.5 | 87.4 | 49.2 | 1.5 | 76.2 |
| GPT ⊗ Llama CoLA-Prompt | 69.7 | 1.5 | 88.6 | 64.7 | 1.5 | 84.9 |
| Llama ⊗ Llama CoLA-SFT | 70.0 | 1.5 | 92.6 | 68.9 | 1.5 | 90.4 |
| Llama ⊗ Llama CoLA-RL ens | **75.0** | 1.4 | **94.9** | **71.0** | 1.4 | **91.6** |

Table 5: Human Analysis on QA domains. CoLa - Guide ⊗ Reasoner.

| Strategy | Human ⊗ Llama | | Llama ⊗ Llama CoLA-Prompt | | Llama ⊗ Llama CoLA-RL ens | |
|---|---|---|---|---|---|---|
| | Freq% | Acc% | Freq% | Acc% | Freq% | Acc% |
| *correcting step in a reasoning chain* | 11.7 | 57.1 | 68.4 | 48.1 | 39.7 | 80.0 |
| *propagating reasoner's response leading to contradiction* | 30.0 | 80.6 | 19.0 | 53.3 | 46.0 | 69.0 |
| *asking the original question based on new context* | 26.7 | 68.8 | - | - | - | - |
| *asking for justification of unsupported claim* | 7.5 | 77.8 | 1.3 | 100.0 | 3.2 | 0.0 |
| *clarifying quantifier* | 5.0 | 66.7 | 2.5 | 0.0 | 3.2 | 50.0 |
| *asking to focus only on question* | 0.8 | 100.0 | 2.5 | 50.0 | 4.8 | 100.0 |
| *asking word sense disambiguation* | 1.7 | 50.0 | 5.1 | 100.0 | - | - |
| *clarifying order of sequence* | 2.5 | 66.7 | 1.3 | 100.0 | 1.6 | 100.0 |
| *asking to enumerate all possible options* | 4.2 | 80.0 | - | - | - | - |
| *asking to compare options* | 4.2 | 100.0 | - | - | - | - |
| *formulating statement as question* | 4.2 | 60.0 | - | - | - | - |
| *referencing to the option required to solve the problem* | 0.8 | 100.0 | - | - | 1.6 | 100.0 |
| *metaphorical thinking* | 0.8 | 100.0 | - | - | - | - |
| **Total Interactions** | **120** | | **79** | | **63** | |

Table 6: Strategy distribution for different Guides on CSQA2 Human Analysis

interaction examples. Reflecting on these limitations we think it would be an interesting future work to design general purpose guides, which can be paired with any LLM as the collaborative partner, without additional supervision.

