# OpenReview forum: "CoLa: Learning to Interactively Collaborate with Large Language Models"
_colmweb.org/COLM/2025/Conference — COLM 2025_

### Official Review · Reviewer_xE3z · 2025-05-10

**Rating:** 7
**Confidence:** 3
**Ethics Flag:** 1

**Summary:**

In this paper, the authors investigate collaborative problem solving involving two models, a reasoner (e.g., a frozen LLM that directly solves tasks) and guide model (i.e., any agent or model providing feedback to the reasoning model), with the aim of mimicking the kinds of human-machine interactions studied in HCI. They specifically ask (line 30): “can LLMs simulate the role of humans in these interactions [i.e., these guide models that provide feedback to the reasoners]?” (I was initially very confused by their described aim of “simulating humans” expressing in line 30, with the thoughts that all language models to some degree simulate humans, but this confusion was made more clear by section 3. What is still left clear is if their framing on line 131 is in any way different from a standard chat format, for which we have loads of publicly available and models).

To accomplish this, they propose the following processing pipeline (nicely summarized in Figure 1): collect a small number of seed interactions from humans (I initially thought that this involved significant human annotation, but this is just about getting in-context examples; they don't say who the human are here, is it the authors?); 2) use this seed data as in-context examples to elicit human-like feedback data from chatGPT; 3) filter these dialogues for end task correctness (i.e., checking whether the interactions between these two models terminated in the reasoning answer predicting the correct answer) ; 4) use this filtered “clean” data to train a series of SFT and reward/RL models resulting in a final “guide” model (these models are all small llama models). This guide model is then coupled with a large frozen LLM reasoner (ChatGPT 4o-mini) in a dialogue loop and evaluated on 2 QA datasets (commonsenseQA 2 and socialIQA), a puzzle dataset (NYT Connections) and a constrained generation task (CommonGen).

They compare against a comprehensive set of baselines: two different CoT approaches, self-refine (i.e., an approach with a self critique and revision components both performed by a single model) and multi-agent debate (MAD) (i.e., an approach involving multiple agents debating their responses with the aim of arriving at a consensus). The results are impressive: their full approach with a small RL guide model outperforms all approaches, and has sizable gains over ablations of the guide learning components (e.g., removing the RL or SFT training altogether, or just using a prompt-based GPT model. In the latter case, this also does well, but is still outperformed by their full approach, which has the advantage of relying on a much smaller model and hence incurring (probably) considerably less inference costs). Particularly intriguing is the improving over MAD and the self-refine, showing that their particular style of multi-agent interaction is effective.

**Questions To Authors:**

-  did you consider training a guide model on all task data combined?  What’s the advantage of training to each specific task?

- Did you look at the generalization ability of the guide models across tasks?

- A natural ablation involving your chat data would be to compare against a model trained on, say, ultrachat or some other dataset of the same form. Did you consider this? The same for the reward models; can you just use an off-the-shelf reward model?

- I’m assuming that for the llama 3 models, you are using the instruction tuned version of these models, is this correct?

**Reasons To Accept:**

- A novel interactive prompting strategy that mimics human-assistant interaction, one that I could imagine others employing in their work.

- Compelling empirical evidence of its effectiveness compared against other dialogue- and revision-based approaches (self refined, MAD) (some caveats to this below); could be potentially used as a replacement for these approaches.

- A new training protocol for tuning SFT/RL models from distilled GPT dialogue data.

**Reasons To Reject:**

- (minor) I really don’t understand the section 3.3.3 about the “ensemble design”, even after a few times reading it. Specifically, it’s hard to tell if this is just an implementation detail or something significant about their modeling. Please address this.

- While there are clear gains over self-refine and MAD, it is worth bearing in mind that their  “guide” models (and the auxiliary reward models) are specially tuned on each target dataset. Therefore, one should be careful to say that their particular strategy is in general better than these standard refinement strategy, which lack this extra task-specific tuning. I would like to see the authors address this and answer the questions I list below. Particularly, I’m left wondering how general these “guide” models are outside of these narrow tasks.

- As I noted above, I question whether it is fair to claim that your guidance model “simulate the role of humans in these [problem solve] interactions” in a way that’s significantly different than any other form of model training on chat data. There are no ablations that address this (see question below). There are also no ablations on the reward models being used.

---

> ### Author Response · Authors · 2025-06-01
> **Author Response (3)**
>
> ### **Clarifications**:
> - The seed examples are collected by the authors and other human participants, and we can add this in the paper.
> - Yes we use the instruct models with chat capability “Llama-3.2-3B-Instruct” and “Llama-3.2-1B-Instruct”.
>
> ---
>
> We hope these responses resolve the outstanding concerns and demonstrate the contributions and general value of the CoLa framework. Thank you again for helping us improve the paper and we would be more than happy to address any further questions!

---

> > ### Author Response · Authors · 2025-06-05
> > **Follow-up**
> >
> > Dear Reviewer,
> >
> > Given the response period is mid-way we are following-up if our responses addressed your comments? If yes, then would you please consider raising the score?
> >
> > Thanks,
> > Authors of CoLa.

---

> ### Author Response · Authors · 2025-06-01
> **Author Response (2)**
>
> ### **Ensemble design**
> We believe that **our ensemble offers a research contribution**, rather than an implementation detail. It is applicable to many other scenarios, beyond our specific work. Any LLM-based reasoning approach can leverage the ensemble design, and it enhances and generalizes *RL-AI-F* approaches (i.e., using a self-supervised reward function for RL training of an LLM).  Intuitively, the ensemble idea leverages the fact that each instance can be solved using multiple “experts”, i.e., Guide-Reasoner interactions leading to a solution. These interactions could result in different solutions to the problem.
>
> We suggest an **EM-based approach**, where in the E-step we generate solutions from guided-interactions and score them using a reward function.
> Our main insight, the “*wisdom of the crowds*” principle can be utilized to reduce the variance problem of the model when performing the M step. We calculate the MAP solution for the problem, by calculating the weight average reward for each final solution step, and use it in the M-step where the parameters of the reward function are updated.
>
> The **high-level algorithm** of the ensemble design is as follows:
> - During the start of the conversation we prompt the Reasoner to generate top-`K` responses. Each response is then provided to the Guide and the interactions are run in-parallel, creating `K` conversations thereafter.
> - The rationale is that we generate `K`-fold conversations for each problem, resulting in multiple rationales as solutions to the original problem, such that the answer *label* can be estimated through ensemble (**Eq. 3**, pg. 6).
> - The estimated answer *label* provides training data for the Critique:
>   - `+` data: all the conversations which concluded the estimated *label*
>   - `-` data: all the other conversations
> Therefore, with every iteration improving the critique yields “better” reward signal for the Guide
> - We use `K=5` for SocialIQA and `K=3` for CSQA2.

---

> > ### Comment · Reviewer_xE3z · 2025-06-06
> >
> > Great, this is indeed clarifying. Please add these more details into the paper to make it easier to understand.

---

> > > ### Author Response · Authors · 2025-06-07
> > >
> > > We are very glad that our novel ensemble design is now clarified! Sure we will add this explanation in the paper.

---

> ### Author Response · Authors · 2025-06-01
> **Author Response (1)**
>
> Thank you so much for the detailed reading to provide a comprehensive review and for raising several clarifying questions!
> ### **CoLa vs Chat LMs**
> In general, we agree that the tension between general-purpose guides and task-specific guides deserves careful study and this is very much in our plans for future work.
> However, our suggested settings are very different from existing chat/interactions scenarios and datasets studied in the past (such as UltraChat [1], Alpaca [2], CAMEL [3], etc.), whose turns are open-domain, non-goal-directed, and do not emphasize “scaffolding” moves that steer the LLM in an intentional way (e.g., *point out flaws in their reasoning*, *clarify intent*, etc.).
> We believe that obtaining human demonstrations for more generalized intentional reasoning steps across multiple tasks at a large scale deserves its own focused work. Our goal in this paper is to establish the framework that would enable that study.
>
> Our goal is not to claim that LLMs in general fail to “simulate humans,” but rather that the specific social role a human adopts in collaborative problem solving – **a Guide who questions, clarifies, and corrects a partner** – is absent from ordinary chat corpora. As a result, we suggest a **Guide module that is driven by human demonstrations and adapts to the abilities of the Reasoner**. We conduct qualitative evaluation comparing human and CoLa guides, both in terms of their performance (Tab. 6, pg. 19) and guidance strategies used (Fig. 3, pg. 19 and Tab. 7, pg. 20), to study the **interactive collaboration for goal oriented domains**.
>
> ---
>
> ### **Generalization**
> We agree that as part of studying general-purpose vs. task-specific guidance, the comparison with prompting methods that rely on general purpose LLMs should be clarified. We do not claim that our approach is directly comparable with MAD and self-refine, since we rely on self-supervision and (some) gold labels in addition to human demonstrations. This supervision is needed to accomplish our goal — aligning the guidance steps with the LLM reasoner abilities.
> Please note, that **our task adaptation only refers to the guidance steps**, the **LLM reasoner is always general-purpose and not fine-tuned**. The general-purpose version of CoLa which uses a GPT-4o for guidance similar to MAD and self-refine underlying LLM, provides **results that are comparable with MAD and self-refine** (performing slightly worse or better, depending on the task). Our main takeaway from the comparison with MAD and self-refine, is to emphasize the point that **interaction with data is needed in order to push improvements on human-guidance demonstration** as it takes into account the outcomes of those guidance strategies.
>
> ---
>
> ### **Critique Ablation**
> To address the ablation that uses an off-the-shelf reward model, we perform **Critique Analysis** (Sec. 4.3); experimental details are given in App. A.3 (pg. 18). The results are summarized in Tab. 5 (pg. 19):
> - **Off-the-shelf reward model** — on average, the off-the-shelf *Critique* assigns higher positive feedback to responses generated by the **CoLa Reasoner** than to those produced by the **CoT Reasoner** LLM.
> - **Trained reward model** — when we train the reward model with additional supervision—SFT and RL ensemble critiques—it develops stronger critiquing ability, assigning higher positive feedback to conversations that match the correct ground-truth label.
>
>
> ---
>
> ### References
>
> [1] UltraChat: Enhancing Chat Language Models by Scaling High-quality Instructional Conversations, Ding et. al., EMNLP 2023.
>
> [2] Alpaca: A Strong, Replicable Instruction-Following Model, Taori et. al., [https://crfm.stanford.edu/2023/03/13/alpaca.html](https://crfm.stanford.edu/2023/03/13/alpaca.html).
>
> [3] CAMEL: Communicative Agents for "Mind" Exploration of Large Language Model Society, Li et. al., NeurIPS 2023.

---

> > ### Comment · Reviewer_xE3z · 2025-06-06
> >
> > > However, our suggested settings are very different from existing chat/interactions scenarios and datasets studied in the past (such as UltraChat [1], Alpaca [2], CAMEL [3], etc.), whose turns are open-domain, non-goal-directed, and do not emphasize “scaffolding” moves
> >
> > I think intuitively that this is a fair point, but absent of qualitative analysis it's hard to know if indeed these resources are more "open domain", "non-goal-directed" as you suggest. Anyways, this is a minor point, I would just ask that you explain this a bit better in a future draft.
> >
> > > We do not claim that our approach is directly comparable with MAD and self-refine, since we rely on self-supervision and (some) gold labels in addition to human demonstrations
> >
> > Good. please calrify in the future draft.
> >
> > > Critique Ablation
> >
> > Thank you for this extra information, it does addres some of my questions.

---

> > > ### Author Response · Authors · 2025-06-07
> > > **Acknowledgement**
> > >
> > > Thank you so much for responding and providing valuable suggestions to improve our draft! We are glad that you found our responses as helpful and clarifying.
> > >
> > > > Explaining the nature of interaction difference between chat models vs CoLa.
> > >
> > > Yes we will add this explanation in our next draft version.
> > >
> > > > Clarifying the direct comparison with MAD and self-refine.
> > >
> > > Sure, we will add this clarification in Sec 4.1 **Baselines**.
> > >
> > > > Usefulness of Critique Ablation information.
> > >
> > > That's great, we are glad it addresses some of your questions.

---

> > ### Author Response · Authors · 2025-06-07
> > **Thank You**
> >
> > Thanks again for your comprehensive review and engaging in the responses! We are glad that our explanations addressed your comments, and thanks for raising the score!

---

### Official Review · Reviewer_iRuu · 2025-05-11

**Rating:** 6
**Confidence:** 5
**Ethics Flag:** 1

**Summary:**

This paper proposes to develop a model which mimics human in collaborating and guiding LM in solving reasoning tasks. The proposed approach starts from real human interaction with strong LMs like GPT-4, then collect these seed conversations for few-shot examples to collect more conversations. The guide model is then fine-tuned with these data, and further refined via RL. Experiments show the effectiveness of this approach compared with commonly used baselines.

**Questions To Authors:**

The CoLa framework starts from human-LM interactions, and should be critically depends on the quality of the seed demonstrates. I wonder how the final model performance correlates with the length of conversations between human and model? The way human interacts with the model can be very subjective, it can guide the model for 10 steps with simple instructions, or just 2 steps with heavy and detailed instructions. It would be great to define a criterion or structure for human guidance, such that LM can solve challenges problems with minimal (but necessary) human guidance.

**Reasons To Accept:**

This paper developed a principled way of developing guidance models to help LMs solve problems more efficiently. This distillation procedure from human demonstrates to a guidance model is demonstrated to be effective.

**Reasons To Reject:**

The proposed approach however, is not that much different from existing agent modeling techniques which first makes or revises plans at each LM reasoning steps. They both behave like meta-planners that guide the LM to achieve solutions step-by-step. Therefore the novelty of the proposal is limited, and the baselines for comparisons are not strong enough.

---

> ### Author Response · Authors · 2025-06-01
> **Author Response (3)**
>
> ### **Response to Questions**:
> We agree that the way humans approach problem solving is subjective and could result in different interaction demonstrations for the same problem instance. However, the suggested “steps-to-completion” criteria is not directly applicable in our settings, as the demonstrations do not consist of a static set of simpler reasoning steps to be performed by the Reasoner. Rather, it is a rich goal oriented dialogue exchange between the Guide and Reasoner. The steps consist of different guidance strategies, which are applied dynamically to steer the Reasoner to a better solution (e.g., *point out contradictions*, etc.).
>
> Accounting for the diversity of human guidance in these settings is subject to further study. To help mitigate it, we asked human guides to provide their insights to multiple problems, and set a threshold on the maximum number of interactions in these demonstrations, as `max steps` (Tab. 4 pg. 15), discussing the entire process in Appendix A.1 (pg. 13). We used multiple sets of human demonstrations when generating the few-shot GPT-4o demonstrations to new instances, which allowed our Guide module to incorporate the diverse interaction choices.
>
> Having said that, we agree that there is no single "best” guidance interaction, and this problem deserves significantly more study (incorporating both human aspects typically studied in the context of HCI and human decision making, and  the Reasoner’s capability to utilize that guidance step effectively). Our goal in this paper is to introduce the framework that would enable that study.
>
> ---
>
> We hope these responses address your concerns and demonstrate that CoLa contributes a fresh perspective on human-style collaboration rather than incremental prompt engineering. Thank you again for your constructive feedback and we will be very happy to address any further questions or comments you may have.

---

> > ### Author Response · Authors · 2025-06-05
> > **Follow-up**
> >
> > Dear Reviewer,
> >
> > Given the response period is mid-way we are following-up if our responses addressed your comments? If yes, then would you please consider raising the score?
> >
> > Thanks,
> > Authors of CoLa.

---

> ### Author Response · Authors · 2025-06-01
> **Author Response (2)**
>
> ### **Example Interaction (SocialIQA)**
> This example demonstrates the interactive nature of collaboration between the Guide and Reasoner.
> ```text
> Guide: Skylar took Taylor's dog to the vet for his shots without asking. Question: What will happen to Skylar?
> A. be reimbursed for the shots
> B. yell at Skylar
> C. be upset
> Let's think step by step.
>
> Reasoner: Skylar took Taylor's dog … the most likely outcome is that Taylor will react negatively to Skylar's actions. The correct answer is B.
>
> Guide: This is an invalid choice because how can Skylar yell at Skylar? Think about which answer choice is most valid for what would happen to Skylar.
>
> Reasoner: You're right … Considering the situation: … Taylor might be upset with Skylar for not asking permission. So, the most valid answer that reflects what will happen to Skylar is: C.
>
> Guide: But why would Skylar be upset, nothing bad happened to her? She took the dog without informing Taylor but maybe she did the right thing because the dog would have needed the shots, so what will happen next?
>
> Reasoner: You're correct … Taylor might recognize the good deed despite the lack of permission. Therefore, what will likely happen next is: A.
>
> Guide: What is the final answer A, B, or C? Provide the answer in \Answer{}. Eg, \Answer{A}.
>
> Reasoner: \Answer{A}.
> ```

---

> ### Author Response · Authors · 2025-06-01
> **Author Response (1)**
>
> Thank you so much for your time going through our work and providing feedback!
> We would like to clarify how our work differs from existing agent architectures, such as the Plan and Refine style of works and highlight the novelty of our work.
> ### **Novelty**
> At a high-level, the goal of our work is to study how to **simulate human-machine interaction** when collaborating on complex tasks (e.g., a human providing feedback to LLM reasoners to improve the writing of an essay, etc.). Our key insight is that humans employ critical thinking strategies, to push the LLM towards better  task completion based on their understanding-of, and intuition-about, the task. However, this suffer from two key drawbacks - first, it is still a laborious process (e.g., when the task needs to be performed at scale), and second, humans could find it challenging to adapt their guidance strategies to the abilities of the Reasoner as it requires careful examination of the Reasoner module at scale.
> As a result, we suggest a **Guide module that is driven by human demonstrations** and **adapts to the abilities of the Reasoner**. We suggest a novel ensemble-based reward module for using RL in this adaptation process. We conduct qualitative evaluation comparing human and CoLa guides, both in terms of their performance (Tab. 6, pg. 19) and guidance strategies used (Fig. 3, pg. 19 and Tab. 7, pg. 20).
>
> ---
>
> ### **CoLa vs Plan and Revise architectures**
> Although this is not our main objective, we believe that our framework is also novel compared to other agent based approaches which decompose complex problems, for the following reasons:
> - **CoLa supports interactive collaboration between agents** — **Guide and Reasoner through dialogue acts not plans**, and we draw this design inspiration from **human-machine collaboration**, by simulating human behavior by proposing a learning paradigm.
>   - **The interaction steps are not part of a plan** to be executed (i.e., problem simplification through decomposition), but rather correspond to **human efforts in steering an LLM Reasoner** to perform the task better. These include *identifying contradictions* in the Reasoner's steps, *correcting mistakes*, *clarifying usage of quantifiers*, etc.  We identify repeating guidance strategies, both for humans and CoLa Guide,  and conduct an analysis comparing strategy employed by humans and CoLa guides (Fig. 3, pg. 19) and their strategy’s correlation with successful task completion (Tab. 7, pg. 20).
> - We agree that CoLa has some resemblance to other agent architectures, which use an LLM solver, guided by a high-level LLM planning module (e.g., Agent Lumos [1]). However, we note two **key differences** — as previously mentioned:
>   - The CoLa Guide uses a wide range of guidance strategies, not just problem decomposition.
>   - CoLa operates in an interactive fashion, where the guidance strategies are dynamically decided based on the Reasoner’s previous and latest response(s).
> This contrasts with most work (Least to Most Prompting [2]; Plan and Solve Prompting [3]; ReAct [4]; ReWOO [5]) in which the planning module generates a multi-step plan for a powerful LLM or external API to complete. Furthermore, these approaches are prone to failure when any intermediate step results in an error, as cascading errors will be propagated to the final solution step.
> - The second line of work which iterative refines LM’s response by either using the same LLM to generate the response and feedback or multiple agents to refine the response (Self-refine [6]; RL4F [7]), have demonstrated this ability only present in large LMs. Whereas CoLa refines the LM’s response and pushes task performance for both large as well as small LMs, without alternating the LM parameters.
>
> ---
>
> ### References
>
> [1] Agent Lumos: Unified and Modular Training for Open-Source Language Agents, Yin et. al., ACL 2024.
>
> [2] Least-to-Most Prompting Enables Complex Reasoning in Large Language Models, Zhou et. al., ICLR 2023.
>
> [3] Plan-and-Solve Prompting: Improving Zero-Shot Chain-of-Thought Reasoning by Large Language Models, Wang et. al., ACL 2023.
>
> [4] ReAct: Synergizing Reasoning and Acting in Language Models, Yao et. al., ICLR 2023.
>
> [5] ReWOO: Decoupling Reasoning from Observations for Efficient Augmented Language Models, Xu et. al., arXiv 2023.
>
> [6] Self-Refine: Iterative Refinement with Self-Feedback, Madaan et. al., NeurIPS 2023.
>
> [7] RL4F: Generating Natural Language Feedback with Reinforcement Learning for Repairing Model Outputs, Afra Feyza Akyurek et. al., ACL 2023.

---

> ### Author Response · Authors · 2025-06-07
> **Follow-up**
>
> Dear Reviewer,
>
> The discussion period is about to end in 3 days, so we are following-up if you had the chance to review our responses — which highlight the novelty of CoLa and answer your other comments.
> In particular, we'd love to understand if our response helped address your concerns around the novelty of CoLa compared to existing plan-and-revise agent architectures. We’ve aimed to emphasize that CoLa models interaction through dialog acts (not static plans), explicitly simulating human guidance behavior that adapts based on the Reasoner’s responses — a setting not directly addressed by meta-planners or self-refinement pipelines.
>
>
> We would really appreciate your thoughts and would love to answer any follow-up questions you may have, and we greatly appreciate your time and effort in reviewing our work.
> If you think that our responses address your comments then would you please consider raising the score?
>
> Thanks,
> Authors of CoLa.

---

> ### Comment · Reviewer_iRuu · 2025-06-07
>
> Thank you to the authors for the detailed response, it addressed several of my concerns. Upon further reflection, I agree that designing the agent workflow from an HCI perspective could serve as a key differentiator from existing agent frameworks. I recommend emphasizing this aspect more clearly as the primary motivation behind the proposal. Accordingly, I have updated my rating to 7.
>
> That said, I would still encourage the paper to include a more in-depth discussion of the strengths and limitations of the proposed HCI-driven agent reasoning approach. Practitioners looking to adopt this framework will face immediate challenges in designing effective human-LM interaction protocols. In particular, differing design paradigms could lead to significant variability in reasoning performance. It would be great to provide some guidelines and recommendations for researchers to consider.

---

> > ### Author Response · Authors · 2025-06-07
> >
> > Thank you so much for following-up and we are glad that our explanations helped to demonstrate the novelty of CoLa, from the lens of HCI differentiating from meta-planners style of architectures. Thank you for raising the score to 7!
> > > Accordingly, I have updated my rating to 7.
> >
> > **We notice that the Rating score in the form is 6, so could you also please update that?**
> >
> > Response:
> > - Thanks for making the suggestions to help us improve the paper! We discuss multiple frameworks and architectures in the Related Work section, including HCI. We can expand on the strengths and weaknesses of some of these works, in the paper.
> > - To address variability, we discuss the guidelines and design decisions in **A.1 Seed Set Conversation Generation** (pg. 13). But it is a good suggestion to include general guidelines for any task, while designing the high-level pipeline to build human interaction.
> >
> > Thanks again for your feedback and we would really appreciate if you can please update the score in the form!

---

> ### Author Response · Authors · 2025-06-08
> **Requesting to Update Score Mismatch**
>
> Dear Reviewer,
>
> Thanks again for responding and raising the score! We notice a score mismatch between the comments (7) and the **Rating** (6) in the form:
> > Accordingly, I have updated my rating to 7.
>
> Can you please update the **Rating score** as well? We would really appreciate your timely update as the response period is ending soon!
>
> Thank you so much,
> Authors of CoLa.

---

> > ### Author Response · Authors · 2025-06-09
> >
> > Dear Reviewer,
> >
> > Following up, if you can please fix the **Rating score** in the form based on your comments.
> > > Accordingly, I have updated my rating to 7.
> >
> > The response period is ending in 1 day, so will really appreciate your timely update! Thanks again for your time!
> >
> > Thank you,
> > Authors of CoLa.

---

> ### Author Response · Authors · 2025-06-10
>
> Dear Reviewer,
>
> We noticed that the **Rating score** is still **6**, so can you please update that to **7** based on your comments:
>
> > Accordingly, I have updated my rating to 7.
>
> The rebuttal period is ending **soon**, so we would be very grateful if you can please fix the main score timely!
>
> Thanks a lot,
> Authors of CoLa.

---

### Official Review · Reviewer_zghj · 2025-05-13

**Rating:** 8
**Confidence:** 4
**Ethics Flag:** 1

**Summary:**

This paper addresses the challenge of mimicking human behavior when two agents work together to solve complex, multi-step tasks. While this is an important and difficult problem, the authors suggest that in goal-oriented dialogue settings with a clear focus, such behavior can be approximated quite effectively. The goal of this work is to model the interaction between two agents: a Reasoner and a Guide. The Reasoner is a language model that processes and generates information, while the Guide is designed to imitate a human participant in a dialogue (also implemented as a language model). The current approach focuses on training the language model acting as the Guide. The authors argue that, after training a relatively small language model, it performs better as a Guide than a more powerful language model. They also argue that this is true whether the Reasoner is a small or a large language model.

Two training strategies are used in this work: fine-tuning and reinforcement learning (RL). The main difference between them is that fine-tuning involves training models on data where each conversation leads to the correct solution, while RL is conducted on data where conversations may result in either correct or incorrect answers. Each dialogue act generated is assigned a measure: if it leads to the correct answer, +1 is added to the count, otherwise -1 is added to it. This is used to fine-tune the model, called the Critique model. Additionally, the paper introduces an ensemble approach for to further train the Critique model. Instead of generating a single turn at each dialogue step, the model produces K-top outputs of the beam search, which increases the diversity and the sheer number of conversations, and thus results in an expanded training set. The evaluation is conducted for four datasets: two question-answering (QA) dataset, a puzzle dataset, and a constraint generation dataset. For the QA and puzzle tasks, performance is measured against ground truth labels. In the constraint generation task, the performance is evaluated against human performance (i.e., human written texts), with GPT-4 serving as the judge. The authors also report conducting human experiments to compare human performance to their models for the question-answering tasks.

The findings are interesting. One of them is that performance drops when a powerful Reasoner is combined with a smaller Guide model. For both smaller and larger Guides, RL boosts performance. The authors claim that humans usually give information in advance before asking a question in QA tasks, but this approach is absent in their models, which is a very interesting finding, given that in some other aspects, the models do really well, e.g. their RL model does better than humans in correcting intermediate reasoning steps.

**Reasons To Accept:**

The paper reports a few interesting design approaches to training and data generation for the training.
The paper shows comprehensive evaluation across multiple datasets, including human performance benchmarks.
Human experiments are conducted to analyze against them behavior of the models.
The topic of the paper is interesting and challenging.

**Reasons To Reject:**

None

---

> ### Author Response · Authors · 2025-06-01
> **Author Response**
>
> Thank you so much for your time for carefully reviewing our work and acknowledging the contributions. We would like to express our gratitude for the positive feedback! We are glad that you found the topic of this paper interesting and challenging!
>
> We are delighted that our main contributions resonated with you. To reiterate the points you highlighted:
> - We introduce CoLa, which learns a Guide agent that emulates human steering and collaborates with a fixed Reasoner agent, enabling effective multi-step problem solving.
> - We present a three-stage training pipeline—seed human demonstrations, supervised fine-tuning, and reinforcement learning with a learned Critique—that yields high-quality, strategy-rich conversational data for the Guide.
> - CoLa is benchmarked on two commonsense QA datasets, a challenging word-puzzle (NYT Connections), and a constrained text-generation task (CommonGen), demonstrating versatility across interaction domains.
> - After training, a lightweight Llama-3B Guide consistently outperforms a much larger GPT-4o Guide in the same role, demonstrating that training leads to a "better" collaboration.
> - Our Human analysis reveals complementary strategies: humans often front-load additional context that CoLa currently omits, while CoLa excels at spotting and correcting intermediate reasoning errors after RL fine-tuning.
>
> We appreciate your thoughtful recognition of these contributions and look forward to answering any questions or comments you may have.

---

> ### Comment · Reviewer_zghj · 2025-06-06
>
> Thanks for the response, I hope this approach can be scaled to other domains too, given that such interaction can be explored to accomplish various complex tasks that LLMs cannot carry out on their own (a pure prompt-based approach)  but a Guide can help them to achieve the goal.

---

> > ### Author Response · Authors · 2025-06-06
> > **Thank You**
> >
> > Thank you so much for your encouraging feedback! We're equally excited about the potential of CoLa to generalize to other domains—especially for modeling human expertise as interactive guidance for complex, general-purpose tasks where LLMs alone may struggle.

---

### Decision · Program_Chairs · 2025-07-08

**Decision:**

Accept

**Comment:**

Summary
This paper introduces CoLa, a novel framework for training small language models to act as Guides that provide human-like, interactive feedback to a fixed Reasoner model. The Guide is trained via a three-stage pipeline: (1) collecting seed demonstrations from human-LLM interactions, (2) supervised fine-tuning on GPT-augmented dialogue data, and (3) reinforcement learning using a novel ensemble-based Critique model. CoLa is evaluated on four tasks—SocialIQA, CSQA2, NYT Connections, and CommonGen—and consistently outperforms strong prompting and multi-agent baselines like cot, self-refine. The paper also includes a strategy-level comparison between human and model-guided interactions.

Strengths:
- CoLa models human-inspired interactive guidance as a distinct social role in goal-oriented LLM collaboration. Reviewers also said this differentiates it meaningfully from prior agent architectures like plan-execute or self-refine pipelines.
- The system demonstrates performance gains across all evaluated tasks, including outperforming GPT-4-based guides in some cases—despite using much smaller models like LLaMA-3B.
- The ensemble-based Critique design provides a robust reward signal for RL training and generalizes well across instances.
- The paper goes beyond accuracy metrics to compare CoLa’s strategies with those used by human guides, showing complementary strengths, for example CoLa excels in identifying and correcting intermediate errors.

Concerns and Author Responses
1. Novelty was CoLa may be similar to meta-planners or plan-and-revise agents as noted by the reviewers. Authors clarified that CoLa is dialog-driven and dynamic, modeling human feedback strategies unlike static decomposition agents. They framed it as an HCI-inspired collaboration paradigm. We encourage the authors for clearer positioning in the next draft.
2. Unclear if the ensemble was central or an implementation detail, in Section 3.3.3 was difficult to understand; Authors provided a clear breakdown: it functions like an EM-style algorithm, uses top-K guided Reasoner outputs to construct multiple conversational trajectories, and updates the Critique via MAP-style aggregation. This needs to be added to the paper.
3. Authors acknowledged thelimitation, clarified that the Reasoner remains general-purpose, and presented CoLa as a framework rather than a universal guide system.

Despite a few minor residual concerns, the core contributions and methodology of the paper is convincing.